# FABIO: TWAS fine-mapping to prioritize causal genes for binary traits

**Haihan Zhang**[1], **Kevin He**[1], **Zheng Li**[1], **Lam C. Tsoi**[1,2,3], **Xiang Zhou**[1] *

**1** Department of Biostatistics, University of Michigan, Ann Arbor, Michigan, United States of America,
**2** Department of Dermatology, University of Michigan Medical School, Ann Arbor, Michigan, United States of America, **3** Department of Computational Medicine and Bioinformatics, University of Michigan, Ann Arbor, Michigan, United States of America

\* xzhousph@umich.edu

**Data Availability Statement:** The UK Biobank data used in this study were obtained from the UK Biobank resource (http://www.ukbiobank.ac.uk) under Application Number 30686. The GEUVADIS gene expression data used in this study are

## Abstract

Transcriptome-wide association studies (TWAS) have emerged as a powerful tool for identifying gene-trait associations by integrating gene expression mapping studies with genome-wide association studies (GWAS). While most existing TWAS approaches focus on marginal analyses through examining one gene at a time, recent developments in TWAS fine-mapping methods enable the joint modeling of multiple genes to refine the identification of potentially causal ones. However, these fine-mapping methods have primarily focused on modeling quantitative traits and examining local genomic regions, leading to potentially suboptimal performance. Here, we present FABIO, a TWAS fine-mapping method specifically designed for binary traits that is capable of modeling all genes jointly on an entire chromosome. FABIO employs a probit model to directly link the genetically regulated expression (GReX) of genes to binary outcomes while taking into account the GReX correlation among all genes residing on a chromosome. As a result, FABIO effectively controls false discoveries while offering substantial power gains over existing TWAS fine-mapping approaches. We performed extensive simulations to evaluate the performance of FABIO and applied it for in-depth analyses of six binary disease traits in the UK Biobank. In the real datasets, FABIO significantly reduced the size of the causal gene sets by 27.9%-36.9% over existing approaches across traits. Leveraging its improved power, FABIO successfully prioritized multiple potentially causal genes associated with the diseases, including *GATA3* for asthma, *ABCG2* for gout, and *SH2B3* for hypertension. Overall, FABIO represents an effective tool for TWAS fine-mapping of disease traits.

## Author summary

In our study, we developed a new method called FABIO, designed to improve the accuracy of identifying genes linked to diseases. Traditional methods typically analyze genes one at a time, which can miss important connections between genes. FABIO, however, looks at all genes on a chromosome together and focuses specifically on diseases that can be categorized in a 'yes or no' manner, such as asthma or hypertension. By accounting for the relationships between genes, FABIO provides a more precise way to find those that

publicly available at https://www.internationalgenome.org/data-portal/data-collection/geuvadis.

**Funding:** This study was supported by the National Institutes of Health (NIH) grants R01AR080662 and UC2AR081033 (to L.C.T.), and R01HG009124 and R01GM144960 (to X.Z.). The funders had no role in study design, data collection and analysis, decision to publish or preparation of the manuscript.

**Competing interests:** I have read the journal's policy and the authors of this manuscript have the following competing interests: L.C.T has received support from Galderma and Janssen.

may be contributing to the disease. After illustrating the benefits of FABIO through extensive simulations, we tested our method on real data: FABIO reduced the number of potential causal genes by 28–37%, making it easier to pinpoint key genes. For example, FABIO highlighted *GATA3* as important for asthma and *ABCG2* for gout. We believe FABIO will help researchers better understand the genetic basis of diseases and could eventually lead to more targeted treatments.

## Introduction

Transcriptome-wide association studies (TWAS) have gained significant popularity in the field of genetics as they are capable of integrating gene expression mapping studies with genome-wide association studies (GWAS) to uncover potentially causal associations between genes and traits [1–3]. However, the traditional approach of conducting marginal TWAS analyses, which examine one gene at a time, has important limitations. In such analyses, the focus is solely on testing the marginal association between genetically regulated expression (GReX) of a single gene and a trait of interest. This approach overlooks the fact that neighboring genes often exhibit correlated expression levels and share cis-SNPs that are in close linkage disequilibrium (LD) with each other [4,5]. Consequently, genes detected through marginal TWAS may not necessarily be causal but rather represent tag genes in the region. To overcome the limitation of marginal TWAS analysis, several statistical methods for TWAS fine-mapping, including FOCUS (fine-mapping of causal gene sets) [4] and FOGS (Fine-mapping Of Gene Sets) [6], have been developed [7–9]. These TWAS fine-mapping methods can prioritize associated genes by jointly modeling multiple genes within a specific region of interest. The TWAS fine-mapping methods begin by constructing GReX prediction models using data from eQTL mapping studies [2,10]. The constructed prediction models are then used to impute GReX in GWAS, enabling the association testing between GReX and the trait of interest at the locus level. Different TWAS fine-mapping methods employ distinct modeling frameworks. Specifically, FOCUS, a Bayesian method, specifies a sparsity-inducing prior on gene effect sizes on the outcome trait and uses Markov chain Monte Carlo (MCMC) for modeling inference. Conversely, FOGS, a frequentist method, relies on a penalty function to model the gene effect sizes on the outcome trait and applies gene set testing for fine-mapping. TWAS fine-mapping analyses have been carried out in many studies, facilitating the discovery of potentially causal gene-trait associations.

Despite their widespread usage, current TWAS fine-mapping methods have two important limitations. The first limitation is that these methods were primarily designed to model quantitative traits, which are traits that can be measured on a continuous scale, such as height or blood pressure. When applied to binary disease traits, both FOCUS and FOGS still apply a linear regression framework to model the association between multiple GReX and the binary outcome trait (S1 Text), relying on input of marginal GWAS summary statistics generated from either linear regression [11,12] or logistic regression. Such linear modeling approach of TWAS aligns with the common practice in GWAS association analysis, as a linear regression can be viewed as an effective approximation to logistic regression when effect sizes are small [13]. However, unlike the SNP effect sizes in GWAS, the GReX effect sizes observed in TWAS analysis can be relatively large. Consequently, employing linear regression as an approximation to logistic regression in TWAS fine-mapping settings may not accurately capture the binary nature of the outcome, potentially leading to a loss of power. The second limitation of existing TWAS fine-mapping methods is that they only consider the correlation among GReX within

local genomic regions defined by LD blocks. However, cis-SNPs for genes that are outside the same genomic region may still be in LD with each other, resulting in potential correlation among GReX across genomic regions. In addition, correlated gene expression may be observed for pairs of genes located outside local LD blocks, again leading to potential correlation among GReX across genomic regions. Moreover, focusing solely on GReX of genes located within local genomic regions thus may miss causal genes located in other genomic regions that also contribute to the trait of interest [14,15], potentially leading to unnecessarily increase in residual error variance and subsequent loss of power. Therefore, modeling multiple genes on an entire chromosome may be advantageous in capturing GReX correlations that are not captured by modeling small genomic regions individually, thereby potentially increasing the power of TWAS fine-mapping.

Here, we introduce a novel TWAS fine-mapping method called FABIO (Fine-mApping of causal genes for BInary Outcomes) that effectively addresses the aforementioned limitations. FABIO relies on a probit model to directly relate multiple GReX to binary outcome in TWAS fine-mapping. Additionally, it jointly models all genes located on a chromosome to account for the correlation among GReX arising from cis-SNP LD and expression correlation across genomic regions. We demonstrate the effectiveness of FABIO through comprehensive simulations. We also applied FABIO to analyze six binary disease traits in the UK Biobank dataset, where FABIO was able to narrow down a small set of causal genes, with the set size 27.9–36.9% smaller than those yielded by current TWAS fine-mapping methods. Notably, FABIO prioritized multiple candidate genes previously validated for their association with the respective diseases, such as *GATA3* for asthma; *ABCG2* for gout; and *SH2B3* for hypertension. Putative causal genes identified by FABIO were further validated through enrichment analyses, with disease-associated pathways highlighted in asthma and hypertension. These results support the superior power of FABIO in prioritizing putatively causal genes associated with binary traits.

## Results

### Method overview and simulation setup

FABIO is described in the Materials and Methods section, with technical details provided in the S1 Text and a method schematic shown in Fig 1.

Briefly, FABIO is a TWAS fine-mapping method that aims to identify genes whose GReX is associated with a binary trait of interest. Different from existing TWAS fine-mapping approaches, FABIO explicitly models the binary nature of the outcome trait through a probit model, and simultaneously models all genes on an entire chromosome to account for the GReX correlation both within and between LD blocks, which also reduces the residual error variance to enhance the precision of TWAS fine-mapping. We develop an efficient MCMC algorithm for model inference, making FABIO scalable to large biobank scale datasets.

We performed comprehensive simulations to evaluate the performance of FABIO and compared it mainly with two existing TWAS fine-mapping methods FOCUS and FOGS. We also compared FABIO with two recently published methods GIFT and cTWAS in the baseline simulation due to their heavy computational demand. The simulation details are provided in the Materials and Methods section. Briefly, we obtained genotype data from 373 individuals in GEUVADIS and 50,000 individuals in UK Biobank. We focused on 770 genes residing in 50 randomly selected LD blocks on chromosome 1 and obtained *cis*-SNPs for these genes for simulations. For each gene in turn, we used the genotype data of the *cis*-SNPs from GEUVADIS to simulate the gene expression level in the gene expression mapping study. We also used the genotype data from the same set of *cis*-SNPs in UK Biobank to construct the GReX underlying

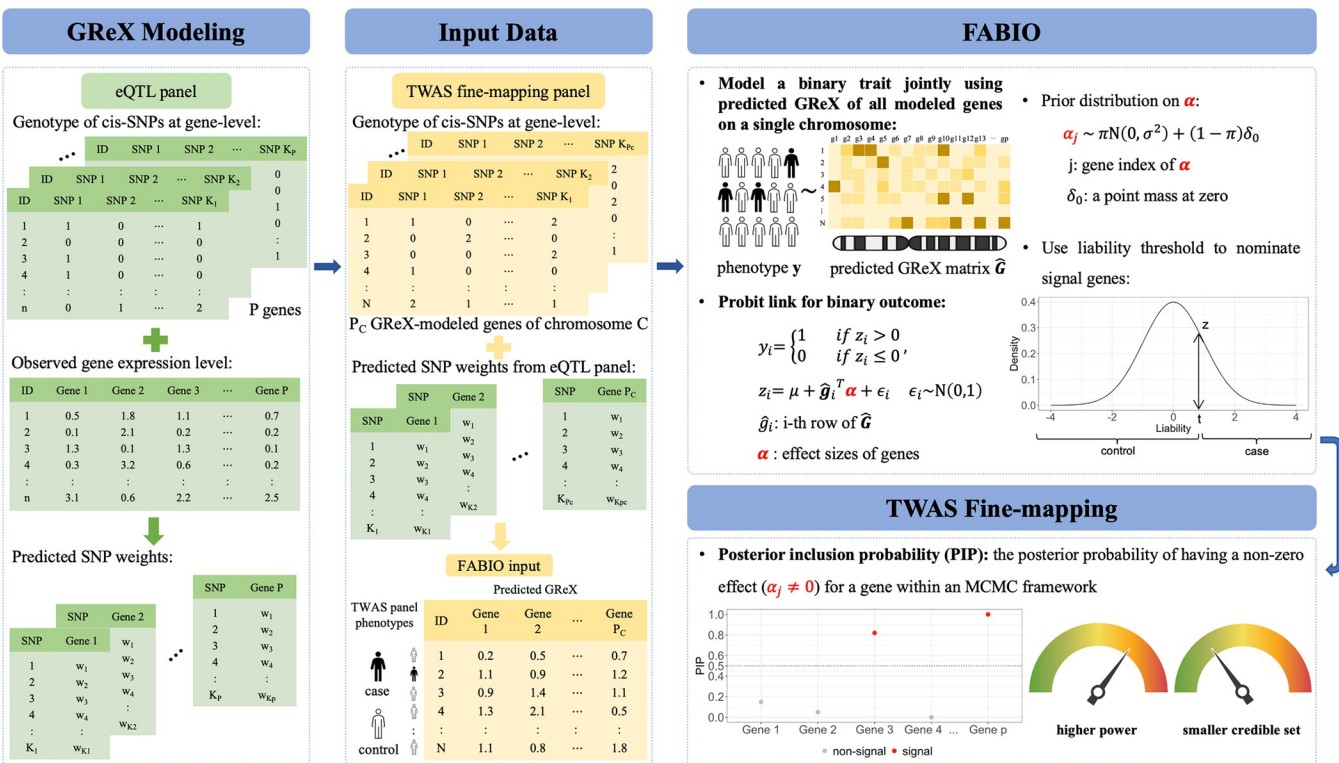

**Fig 1. Schematic overview of FABIO for TWAS fine-mapping of binary outcomes.** Same as other existing methods, as a two-step TWAS fine-mapping approach, FABIO requires predicted GReX of the study cohort generated using the standard method like PrediXcan or BSLMM. Shown under the "GReX Modeling" and "Input Data" section, SNP weights are first estimated from the eQTL mapping cohort (sample size = n) with known genotypes and gene expression levels, then the predicted GReX will be generated in the study cohort (sample size = N, usually N > n), using known genotypes and estimated SNP weights. Shown under the "FABIO" section, FABIO explicitly models the binary nature of the outcome trait through a latent variable **z** with a sparsity inducing prior on each element of the gene effect sizes **α**. It also simultaneously models all genes on a single chromosome to account for the GReX correlation both within and between LD blocks, through the input of individual-level GReX matrix. We apply MCMC method to estimate the model parameters and obtain test statistic for each gene effect size $\alpha_i$, and use the posterior inclusion probability (PIP) as the evidence for the gene's association with the binary outcome trait ("TWAS Fine-mapping" section). * Icons used in this figure are from BioRender.com.

these genes, selected some genes to be causal, and simulated a binary outcome trait of interest in the GWAS. Afterwards, we applied different methods to perform TWAS fine-mapping to identify the causal genes. In the process, we considered a baseline simulation setting and varied one parameter at a time to examine the influence of different parameters on top of the baseline setting. We examined a total of 11 null settings and 15 alternative settings, with 100 simulation replicates per setting.

## FABIO controls false signals in null simulations

First, we examined the performance of different methods under complete null simulation settings where none of the genes is associated with the outcome trait ($PVE_2 = 0\%$). We found that FABIO and FOCUS had reasonably low false positive rates under these settings compared to FOGS (S1A Fig). For example, under the baseline setting, FABIO falsely identified an average of 0.22 genes in the 95% credible set while FOCUS falsely identified an average of 0.35 genes per simulation replicate. In contrast, FOGS produced inflated p-values (S1B Fig) and falsely identified an average of 3.68 genes per replicate under the Bonferroni corrected threshold. Across settings (S1A Fig), FABIO, FOCUS, and FOGS falsely identified an average of 0.25, 0.33, and 4.25 genes per replicate, respectively.

Next, we examined a more challenging null simulation setting where the causal genes were not included in the analysis. Under this setting, because the null genes may display gene expression correlation with the causal genes and may have cis-SNP overlap with them, the null genes will display marginal TWAS signals and could be falsely identified to be causal in the fine-mapping analysis. As expected, we observed much higher false positive rates for all three methods compared to the complete null setting. However, both FABIO and FOCUS still detect much less false signals compared to FOGS. Specifically, with the 95% credible set, FABIO falsely detected an average of 4.43 genes per replicate while FOCUS falsely detected an average of 4.70 genes (S1A Fig). At a Bonferroni corrected threshold, FOGS falsely detected an average of 30.90 genes per replicate (S1A Fig), with highly inflated p-values (S1C Fig).

## FABIO improves power for fine-mapping causal genes in alternative simulations

We examined the performance of different methods in controlling false discovery rate (FDR) under the alternative simulations. Under the baseline setting, the estimated FDR based on PIP for both FABIO and FOCUS are relatively conservative and higher than the true FDR (S1D Fig). In contrast, the estimated FDR based on p-values from FOGS are much lower than the truth, which is consistent with its highly inflated p-values observed in the null simulations and can lead to an excessive number of false discoveries if used in practice. Under an estimated FDR of 0.05, the true FDR is 2.3%, 4.0%, and 45.5% in the baseline setting for FABIO, FOCUS, and FOGS, respectively (Fig 2A). Similar results were observed under different case/control ratios (Fig 2A) and $PVE_1$ (Fig 2B), as well as other simulation settings (S2 Fig): across all settings, the true FDR that corresponds to the estimated FDR of 0.05 is on average 2.8%, 3.4%, and 39.0% for FABIO, FOCUS, and FOGS, respectively. Note that FOGS achieved calibrated type I error in non-sparse settings (S2C Fig), which is consistent with its modeling specifications (S1 Text).

Next, we evaluated the power of different methods in detecting causal genes under alternative settings. Since different methods use different statistics and have different thresholds (e.g., 95% credible set based on PIPs for FABIO and FOCUS and a Bonferroni adjusted p-value threshold for FOGS), we first compared the size of 95% credible set and the number of true signal genes in the 95% credible set between FABIO and FOCUS. Then for fair comparisons among all three methods, we used two rank-based criteria to evaluate method performance: we either computed power based on a true FDR threshold of 0.05 or measured area under the curve (AUC) using the receiver operating characteristics (ROC) curve. In the simulations, we found that FABIO is more powerful than the other two methods under all alternative simulation settings. For example, in the baseline setting, at a true FDR threshold of 0.05, FABIO, FOCUS, and FOGS achieved a power of 54.6%, 36.5%, and 6.8%, respectively (S3A Fig). The power gain brought up by FABIO is partially due to direct modeling of binary traits and partially due to modeling all genes on an entire chromosome together. Specifically, if we restrict FABIO to analyze one LD block at a time same as FOCUS and FOGS, its power is 48.5% (S3A Fig), which is at least 32.9% higher than FOCUS or FOGS, supporting the benefits of explicit modeling of binary traits. If we apply FABIO on an entire chromosome, its power will further increase to 54.6% (S3A Fig), which is at least 49.6% higher than FOCUS or FOGS, supporting the benefits of analyzing all LD blocks on the same chromosome jointly. Measuring AUCs instead of power at FDR produces similar results. Specifically, in the baseline setting, FABIO achieved an AUC of 82.4% and restricting it to analyzing one LD block at a time reduced its AUC to 80.9%. In comparison, FOCUS achieved an AUC of 73.7% (Bootstrap test p-

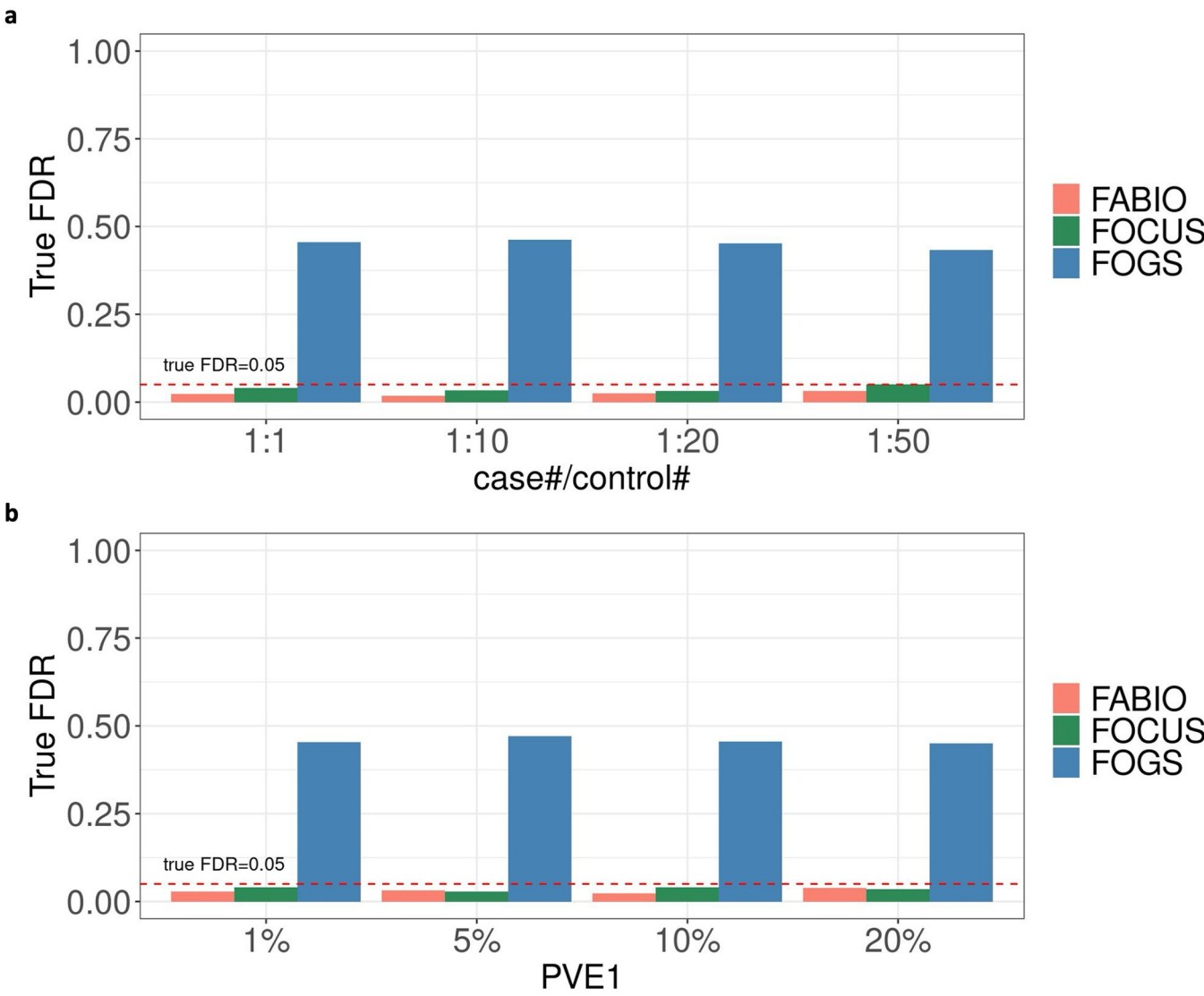

**Fig 2. True FDR at an estimated FDR threshold of 0.05 under different conditions.** The red dashed line indicates a true FDR of 0.05. **(a)** Under different case/control ratios. **(b)** Under different percentages of $PVE_1$.

value = 0.02 for FABIO vs FOCUS) and FOGS achieved 69.0% (p-value = $2.15 \times 10^{-5}$ for FABIO vs FOGS; S3B Fig).

We carefully examined the influence of different simulation parameters on the power of different methods. In terms of case/control ratio, between FABIO and FOCUS, we noticed FABIO can always achieve smaller size of 95% credible set with more true signal genes involved (Fig 3A). We found that the power of different methods all reduced with decreasing case/control ratio, though the rank among them did not change. Specifically, when the case/control ratio decreased from 1:1, 1:10, 1:20, to 1:50, the power of FOGS decreased from 6.8%, 5.2%, 4.2%, to 3.0%; the power of FOCUS decreased from 36.5%, 34.8%, 33.5%, to 23.2%; and the power of FABIO decreased from 54.6%, 50.2%, 49.0%, to 44.8%, which remains higher than the other two methods (Fig 3B). The results of AUCs are similar (Fig 3C). In terms of SNP effects on gene expression ($PVE_1$), we had similar observations on the size of 95% credible set between FABIO and FOCUS (Fig 3D). We also observed that the performance of all

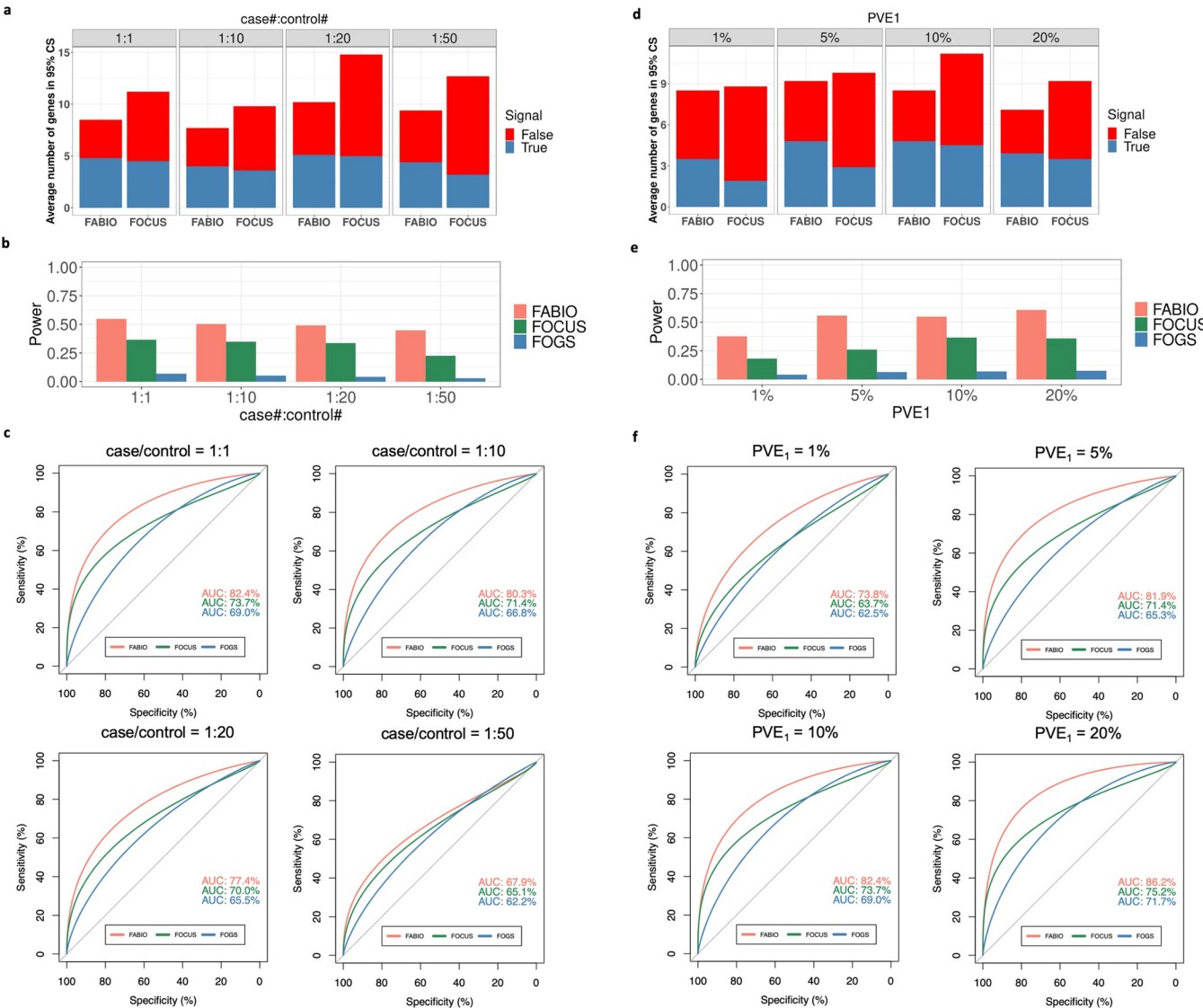

**Fig 3. Performance of different methods in simulations with the change of case/control ratio or PVE1. (a)** Average number of genes in 95% credible set (CS) defined by FABIO or FOCUS and the number of true signal genes in 95% CS under different case/control ratios. **(b)** Power comparison for different methods based on a true false discovery rate (FDR) of 0.05 under different case/control ratios. **(c)** ROC curves of different methods with AUCs recorded under different case/control ratios. **(d)** Average number of genes in 95% credible set (CS) defined by FABIO or FOCUS and the number of true signal genes in 95% CS under different PVE1. **(e)** Power comparison for different methods based on a true false discovery rate (FDR) of 0.05 under different PVE1. **(f)** ROC curves of different methods with AUCs recorded under different PVE1.

methods improved with increasing $PVE_1$, though their rank did not change (Fig 3E and 3F). The power difference between different methods becomes more apparent for relatively small $PVE_1$ (e.g., 5%), highlighting the benefits of FABIO in more challenging settings. For example, when $PVE_1$ = 5%, FABIO still achieves a power of 55.8%, while the power of FOCUS and FOGS reduces to 26.0% and 6.5%, respectively (Fig 3E). In terms of the genetic architecture underlying gene expression, we found that FABIO still identified more true signal genes through 95% credible set than FOCUS (S4A Fig), and all methods had better performance with decreased number of SNPs with non-zero effects on gene expression which is equivalent to increased SNP effect sizes, though their rank again did not change (S4B and S4C Fig). For

example, when the number of causal SNPs for each gene increased from 1% to 10%, although the power of both FABIO and FOCUS reduced substantially (from 54.6% to 22.5% for FABIO; and from 36.5% to 11.8% for FOCUS), FABIO still had twice the power of FOCUS. The power of FOGS reduced slightly (from 6.8% to 4.8%) as expected, given that the power was already low when the number of causal SNPs for each gene is 1% (S4B Fig). In terms of gene effects on the outcome trait ($PVE_2$), similar observations on 95% credible sets between FABIO and FOCUS (S5A Fig), and when $PVE_2$ increases from 0.2% to 0.4% and 0.6%, all three methods achieved significant power increase (from 44.0% to 54.6% and 57.2% for FABIO; from 27.5% to 36.5% and 38.8% for FOCUS; and from 2.5% to 6.8% and 7.2% for FOGS) (S5B Fig) and better AUC (S5C Fig). Finally, the sparsity of causal genes together with causal SNPs also affect method performance. Again, FABIO had better performance than FOCUS in terms of 95% credible sets (S6A Fig), and both methods performed better than FOGS in terms of power and AUC in sparse settings (S6B and S6C Fig). However, FOGS achieved relatively high power (S6B Fig) and AUC (S6C Fig) in non-sparse settings, consistent with its model specification (S1 Text).

We have used a true FDR threshold of 0.05 for the power comparisons described above. However, we note that such true FDR can only be calculated in simulations but unavailable for real datasets. Therefore, we estimated FDR using p-values from FOGS and PIPs from FABIO and FOCUS to further compare the power of different methods based on an estimated FDR threshold of 0.05. For FABIO and FOCUS, the estimated FDR under different settings is conservative and higher than the true FDR; while for FOGS the estimated FDR are lower than the truth, potentially resulting in an excessive number of false discoveries (S1D Fig). Nevertheless, at the estimated FDR threshold of 0.05, FABIO retains the highest power across all 15 alternative simulation settings: the power of FABIO, FOCUS, and FOGS in the baseline setting is 58.5%, 37.4%, and 6.6%, respectively (S7A Fig), and is on average 50.7%, 31.3%, and 5.5% across all alternative settings, respectively. The conclusion holds regardless of the case/control ratio (S7A Fig), the SNP effect on gene expression ($PVE_1$, S7B Fig), the number of causal SNP (s) for each gene (S7C Fig), the gene effect on the outcome trait ($PVE_2$, S7D Fig), and the sparsity of causal genes and SNPs (S7E Fig).

Next, we explored whether the input summary statistics from GWAS marginal analysis would affect the performance of methods. While we used logistic regression to generate marginal GWAS summary statistics as input for FOCUS and FOGS, we found that using linear regression to generate marginal GWAS summary statistics lead to very similar or slightly reduced performance: at a true FDR threshold of 0.05, the power for FOCUS was 33.3% and 36.5%, and for FOGS, 6.0% and 6.8%, for using linear and logistic regression GWAS summary statistics, respectively (S8A Fig). Similar observations are held in terms of AUCs (S8B Fig). In addition, we found that FABIO with the default prior on $\pi$ achieved higher power than using a beta prior, which is influenced by whether prior expectation of $\pi$ matches the truth (S9A and S9B Fig). Finally, FABIO also achieved better performance than GIFT and cTWAS in limited comparisons due to their computational burden (S10A and S10B Fig). These results indicate that FABIO can outperform other existing methods with competitive computing time and retain better performance with smaller GWAS sample size.

## Application to analyzing binary traits in UKBB

We performed TWAS fine-mapping analysis on six binary disease traits through integrating the eQTL mapping study of GEUVADIS and the GWAS of UK Biobank (details in Material and Methods). Specifically, we focused on 14,388 genes residing in 1,433 independent LD blocks and examined their associations with each of the six binary traits. The six binary traits

**Table 1. Summary results of TWAS fine-mapping in UK Biobank.**

| Trait | GWAS risk regions | Significant TWAS genes | TWAS risk regions | Risk regions with GWAS or TWAS signals | FABIO | FOCUS | FOGS |
|---|---|---|---|---|---|---|---|
| AS | 21 | 7 | 6 | 24 | 23 (15) | 56 (10) | 85 (10) |
| BRCA | 15 | 3 | 3 | 18 | 12 (2) | 14 (2) | 65 (4) |
| GO | 18 | 21 | 18 | 33 | 34 (16) | 44 (16) | 81 (12) |
| HT | 175 | 85 | 80 | 228 | 254 (159) | 227 (128) | 231 (117) |
| PRCA | 17 | 2 | 2 | 19 | 21(7) | 18 (1) | 70 (0) |
| RA | 10 | 9 | 4 | 11 | 20 (14) | 24 (13) | 80 (15) |

include asthma (AS), breast cancer (BRCA), gout (GO), hypertension (HT), prostate cancer (PRCA), and rheumatoid arthritis (RA). Among these traits, we identified 256 (232 unique) GWAS risk regions that contain at least one genome-wide significant SNP (p < 5×10$^{-8}$) associated with at least one trait. We performed a TWAS marginal analysis using FUSION and identified a total of 127 (126 unique) significant genes residing in 113 (107 unique) TWAS risk regions across the six traits. Overlapping the results from the two analyses leads to a total of 333 (294 unique) risk regions that contain at least one genome-wide significant SNP or one marginal TWAS significant gene (Table 1). These are the likely risk regions that are more likely to contain true signals than the remaining regions.

The table summarizes the number of discoveries for each of the six disease traits (rows) in the TWAS fine-mapping analysis of UK Biobank. A GWAS risk region (1$^{st}$ column) is defined as an LD block that harbors at least one genome-wide significant SNP (p-value < 5 × 10$^{-8}$). A significant TWAS gene (2$^{nd}$ column) is defined as a gene with a marginal TWAS p-value < 0.05/14,388. A TWAS risk region (3$^{rd}$ column) is defined as an LD block that harbors at least one marginal TWAS significant gene. A risk region with GWAS or TWAS signals (4$^{th}$ column) is defined as an LD block that harbors at least one genome-wide significant SNP or significant TWAS gene. The last three columns list the number of genes discovered by each of the three methods. The number in the bracket is the number of identified genes that are located in a risk region with GWAS or TWAS signals. We used an estimated FDR threshold of 0.05 to declare significance for all methods in the fine-mapping analysis.

Next, we performed TWAS fine-mapping using FABIO, FOCUS and FOGS on all 14,388 genes from the 1,433 independent LD blocks. Same as what we did in the simulations, we estimated FDR using the test statistics generated by different methods and declared significant gene associations based on an estimated FDR threshold of 0.05 for all methods. Note that even though we used the same estimated FDR threshold, such threshold for different methods may still correspond to different levels of false discoveries: as is evident in simulations, the same estimated FDR from FOGS may correspond to a much higher true FDR than that of FABIO or FOCUS. Based on the estimated FDR threshold of 0.05, FABIO identified a total of 364 associated genes (354 unique) from 248 LD blocks across the six traits, with an average of 1.47 associated genes per LD block. These numbers are reduced when a beta prior is used on $\pi$ (S1 Table), since our input parameters of the beta prior may not reflect the unknown truth of the causal gene density. Among the identified genes, 213 of them reside in a risk region with either GWAS or TWAS signals (Table 1). As a comparison, FOCUS identified a total of 383 associated genes (350 unique) from 188 LD blocks across six traits, with an average of 2.04 associated genes per LD block. Among the identified genes, 170 of them reside in a risk region with either GWAS or TWAS signals (Table 1). FOGS identified 612 genes (439 unique) from 263 LD blocks across six traits, with an average of 2.33 associated genes per LD block. Among the identified genes, 158 of them reside in a risk region with either GWAS or TWAS signals (Table 1).

In addition, we found that FABIO identified more gene candidates that lied in the GWAS or TWAS risk regions compared to GIFT and cTWAS (S2 Table). Overall, FABIO was able to narrow down to the smallest set of genes per LD block along with the highest proportion of identified genes residing in likely risk regions in the set (Fig 4A), supporting its superior performance.

We further examined the power and robustness of different methods through down-sampling analysis. Specifically, we applied the three TWAS fine-mapping methods using only half of the samples for the same six traits and examined whether the results obtained in such reduced sample analysis is consistent with full sample analysis. We found that FABIO identified more gene candidates that lied in the GWAS or TWAS risk regions than FOCUS and FOGS in the reduced sample analysis (S3 Table). In addition, FABIO achieved higher consistency between the full sample analysis and reduced sample analysis, with 135 out of 364 (37.1%) genes identified across all six traits in the full sample analysis that were also identified in the reduced sample analysis, more so than FOCUS (86 out of 383; 22.5%) or FOGS (198 out of 612; 32.4%). These observations support the power and robustness of FABIO in the real data application.

We list a few gene examples that are uniquely identified by FABIO. The first example is *GATA3* associated with asthma (PIP = 1), a gene that resides in the known risk regions of asthma (S11 Fig). *GATA3* has been associated with asthma in previous GWAS [16] and acts as a key regulator of T-cell development, Th2 differentiation, and the Th1/Th2 balance to affect asthma risk [17]. Transgenic mice with a dominant-negative mutant of GATA3 display decreased expression level of Th2 cytokines IL-4, IL-5, and IL-13, along with attenuated asthma symptoms, such as airway eosinophilia, mucus production, and IgE synthesis [18], supporting a potentially causal effect of *GATA3* on asthma risks.

The second example is *ABCG2* associated with gout (PIP = 1, Fig 4B). A common polymorphism (rs2231142, Q141K) in *ABCG2* was identified to be associated with gout and gout related symptoms such as serum urate concentration [19–22] in previous GWAS and our GWAS analysis (Fig 4C). ABCG2, the ATP-binding cassette transporter G2, mediates renal and/or extra-renal urate excretion. Consequently, the dysfunction of ABCG2 leads to elevated levels of uric acid, causing hyperuricemia and gout [23,24]. Indeed, ABCG2-knockout mouse has significantly higher serum uric acid level and significantly lower level of urate excretion from the intestine, which are consistent with the observation in patients with ABCG2 dysfunction, indicating that ABCG2 has an important role in extra-renal urate excretion, especially in intestinal urate excretion [25].

The third example is *SH2B3* associated with hypertension (PIP = 1, Fig 4D). SH2B3, also known as lymphocyte adapter protein (LNK), is an intracellular adaptor protein mainly expressed in hematopoietic and endothelial cells [26]. A non-synonymous SNP (rs3184504) in exon 3 of *SH2B3* leads to a substitution from arginine to tryptophan in amino acid 262 (R262W), and was associated with hypertension in previous GWAS [27] and in our GWAS analysis (Fig 4E). Two different mouse models that disrupt LNK signaling further confirmed that LNK regulates blood pressure and renal inflammation [28,29]. One is an LNK deleted mouse model, that represents a complete loss of function of LNK, where hypertension was severely exacerbated with enhanced renal and vascular inflammation compared to wild type animals [28]. The other is an LNK-mutated mouse model, that represents an alteration of function of LNK, where mutation of the Sh2 domain of LNK led to the attenuation of hypertension, albuminuria, and renal inflammation [29]. Both animal models supported that modulation of LNK function in immune cells can regulate blood pressure and the associated inflammation and end-organ damage [26].

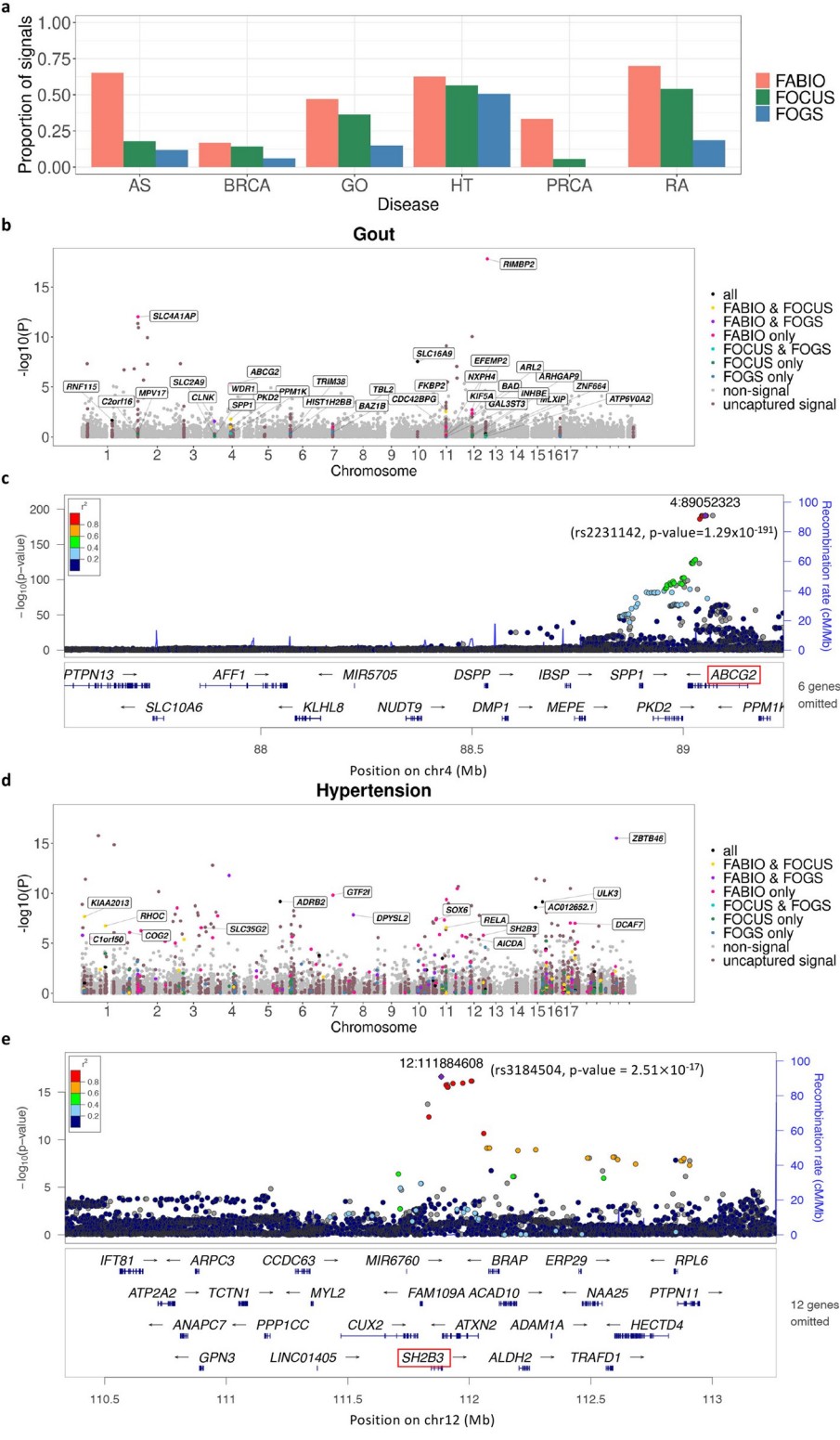

**Fig 4. (a)** Proportion of identified signal genes that locate in risk regions for the six diseases. Highlighted genes identified in TWAS fine-mapping analysis are labeled in black boxes in **(b)** and **(d)**. Each gene is represented as a dot with x-axis indicating its genomic location and y-axis indicating its -log10 of p-value in the marginal TWAS association test. The dot of each gene is then colored based on different categories: (1) only identified by FABIO; (2) only identified by FOCUS; (3) only identified by FOGS; (4) identified by both FABIO and FOCUS; (5) identified by

both FABIO and FOGS; (6) identified by both FOCUS and FOGS; (7) identified by all three methods; (8) located in a known risk region but missed by all three methods (uncaptured signal); (9) a gene shown no significance in both TWAS and GWAS analyses (non-signal). **(b)** TWAS Manhattan plot of gout. **(c)** LocusZoom plot of the corresponding LD block with the GWAS p-value of the SNP rs2231142 in ABCG2 for gout. **(d)** TWAS Manhattan plot of hypertension. **(e)** LocusZoom plot of the corresponding LD block with the GWAS p-value of the SNP rs3184504 in SH2B3 for hypertension.

We further systematically examined the biological relevance of the genes identified by different methods using Gene Ontology and Kyoto Encyclopedia of Genes and Genomes (KEGG) pathway enrichment analyses. For asthma, the candidate genes identified by FABIO are significantly enriched in five pathways, with four from Gene Ontology (S12A Fig) and one from KEGG (S12B Fig). Three of them point to interleukin-13 (IL-13) production: positive regulation of interleukin-13 production (q-value = 0.015), interleukin-13 production (q-value = 0.015), and regulation of interleukin-13 production (q-value = 0.015); and three FABIO significant genes are involved in these pathways, including *TSLP*, *IL4*, and *GATA3* (FABIO PIP = 0.92, 0.76, and 1, respectively). The role of IL-13 signaling in asthma has been recognized for a long time in multiple studies [30–33], and IL-13 is a pleiotropic TH2 cytokine that has been considered as a key to the pathogenesis of asthma through its regulatory role in IgE synthesis, mucus hypersecretion, airway hyperresponsiveness, and fibrosis [31]. A mast cell related pathway, positive regulation of mast cell activation (q-value = 0.015), is also enriched among the candidate genes identified by FABIO, and the involved FABIO significant genes are *SNX4*, *TSLP*, and *IL4* (FABIO PIP = 0.69, 0.92, and 0.76, respectively). Mast cells play key roles in asthma through secretion of mediators with pro-inflammatory and airway-constrictive effects, such as histamine and bioactive lipids [34]. By releasing a variety of such mediators, mast cells can trigger bronchoconstriction and alter the permeability of the bronchial barrier that further lead to asthma symptoms [35]. The inflammatory bowel disease is highlighted (q-value = 0.015) through KEGG enrichment analysis using FABIO significant genes, and the involved genes are *IL18RAP*, *IL4*, *GATA3*, and *SMAD3* (FABIO PIP = 1, 0.76, 1, and 0.75, respectively). A systematic review has shown that asthma is associated with both Crohn's disease and ulcerative colitis, the two types of the inflammatory bowel disease [36]. As a comparison, only two significant pathways (cytokine-cytokine receptor interaction, q-value = 0.006; Thiamine metabolism, q-value = 0.006) were identified from KEGG enrichment analysis using significant genes identified by FOCUS (S12C Fig), and none for FOGS.

We also identified significantly enriched pathways for hypertension. A total of 11 pathways are enriched using the candidate genes identified by FABIO: three from Gene Ontology (S12D Fig) and eight from KEGG (S12E Fig). Two of them are directly associated with blood pressure: regulation of systemic arterial blood pressure (q-value = 0.017) and regulation of blood pressure (q-value = 0.030). Four of them are cardio-related: positive regulation of heart rate (q-value = 0.033), adrenergic signaling in cardiomyocytes (q-value = 0.012), vascular smooth muscle contraction (q-value = 0.025), and dilated cardiomyopathy (q-value = 0.030). Renin secretion (q-value = 0.044) is also enriched, where renin is a well-known regulator in blood pressure through the renin–angiotensin–aldosterone system (RAAS) [37]. Using the candidate genes identified by FOCUS, 17 pathways are enriched, with 14 from Gene Ontology (S12F Fig) and three from KEGG (S12G Fig). However, 10 of them are not directly associated with cardio or blood pressure, such as the top three enriched pathways: positive regulation of adenylate cyclase activity (q-value = 0.024), positive regulation of cyclase activity (q-value = 0.024), and positive regulation of lyase activity (q-value = 0.024). For FOGS, we have two significantly enriched pathways from Gene Ontology (S12H Fig), and one of them is directly related to blood pressure: negative regulation of systemic arterial blood pressure (q-value = 0.026).

Finally, we note that FABIO is reasonably computationally efficient despite the necessary computational cost incurred for direct modeling of binary outcomes. Specifically, in the fine-mapping of UK Biobank data with 337,198 individuals, using eight CPU threads, it took FABIO an average of 16.14 hours to analyze each chromosome per trait. As a comparison, with the same computational resource, it took an average of 11.84 hours for PLINK to compute the GWAS summary statistics to serve as input for FOCUS and FOGS, which incurred an additional 2.52 and 25.08 hours respective for fine-mapping each chromosome. In terms of physical memory, FABIO uses ~52 Gb memory while PLINK uses ~64 Gb memory to compute the necessary GWAS summary statistics for FOCUS or FOGS, which then incurred minimal memory for additional fine-mapping analysis.

## Discussion

In this study, we have presented FABIO, a novel TWAS fine-mapping method tailored for binary outcomes. Unlike existing approaches, FABIO not only incorporates a probit model to capture the binary nature of disease traits but also jointly models all genes on the same chromosome to account for GReX correlation across genes in different LD blocks. As a result, FABIO produces reasonably calibrated test statistics and achieves high power. Through comprehensive simulations and real-world applications to six disease traits in the UK Biobank, we have illustrated the benefits of FABIO.

FABIO employs a probit link function to model the relationship between GReX and binary outcome traits. The probit link function introduces a latent variable known as the liability score, which determines the disease status of each individual. Specifically, when the liability score exceeds a certain threshold, then the individual is classified as having the disease. The liability score is then related to a set of explanatory variables for association analysis. A common alternative approach to modeling binary outcomes is to use the logistic link function within generalized linear models. The logistic link function models the probability of disease for each individual based on the explanatory variables. Consequently, the disease probability varies across individuals depending on the underlying explanation variables. Although the interpretation and inference algorithms differ between these two binary outcome modeling approaches, they often produce similar association results in most application settings [38]. Indeed, in our simulations, we verified that FABIO retains reasonable power even when there are model misspecifications, such as using a logistic regression to generate binary outcomes. The results suggest that FABIO can effectively capture the relationship between GReX and binary outcomes, even when the assumptions of the probit link function are not precisely met.

FABIO accounts for gene expression correlation outside local LD blocks by jointly modeling all genes on the entire chromosome. The constructed GReX indeed displays correlation not only within LD blocks but also between neighboring LD blocks, though the correlations decrease with increasing distance. To illustrate this, we focus on 1,568 genes from 133 distinct LD blocks on chromosome 1. First, we calculated the pairwise correlation of GReX for four distinct groups of gene pairs: those within each LD block, between two adjacent LD blocks, between two distant LD blocks, and between one LD block on chromosome 1 and one on chromosome 2, with the latter serving as a baseline. We observed a clear decreasing trend in GReX correlation across these groups. Notably, the correlations between two adjacent LD blocks and between two distant LD blocks remained substantial—3.88 and 1.47 times higher, respectively, than the baseline correlation between LD blocks on different chromosomes (S13 Fig). This suggests that GReX correlation extends beyond individual LD blocks, although with diminishing magnitude over increasing distance. Second, for all six binary traits examined in our real data analysis, we computed the proportion of phenotype variance ($R^2$) explained by

GReX using linear regression in two ways. First way, we included the GReX for all genes across 133 LD blocks on chromosome 1 in the model and computed the $R^2$. Second way, we included the GReX for all genes within each LD block, computed the $R^2$ for each block, and then summed the $R^2$ values across blocks. Intuitively, if GReX correlation between LD blocks is negligible, the $R^2$ values obtained from these two approaches should be identical. However, we found that the average $R^2$ computed in the first way is 1.38% (±0.49%) while that the average $R^2$ computed in the second way is 1.74% (±0.52%), with a 26.1% increase due to GReX correlation between LD blocks. This highlights the importance of modeling GReX correlation between LD blocks, as is done in FABIO.

FABIO is not without limitations. Firstly, due to the probit link function, FABIO requires individual-level GWAS data as input and cannot directly model GWAS summary statistics. In particular, FABIO needs to infer a latent liability variable for each individual during the MCMC inference process, which is only feasible with individual-level GWAS data. While requiring individual-level data may restrict the application of FABIO, it helps mitigate biases commonly observed in methods that make use of only GWAS summary statistics in the presence of a LD matrix mismatch between the reference panel and the analyzed GWAS data [39,40]. Secondly, similar to FOCUS and FOGS, FABIO adopts a two-stage fitting strategy for TWAS fine-mapping: it relies on a reference eQTL mapping study to construct the gene expression prediction model in the first stage, and applies the SNP weights inferred in the eQTL mapping study to construct GReX in the GWAS data for TWAS fine-mapping in the second stage. While the two-stage fitting strategy is convenient and effective, it overlooks the uncertainty in the gene expression modeling in the first stage, potentially leading to power loss [41,42]. An important future direction is to extend FABIO towards joint likelihood-based inference for TWAS fine-mapping, thus accounting for the inference uncertainty from the first stage. Finally, in our real-world applications, we utilized GEUVADIS data as the eQTL mapping study to fine map the six disease traits. It is important to acknowledge that GEUVADIS data is collected from whole blood, which may not necessary be the disease relevant tissue for the outcome disease. Future extensions of FABIO should consider integrating tissue-specific eQTL mapping studies to reveal tissue specificity underlying the disease trait. Utilizing multiple eQTL panels from various tissues to construct SNP weights has been proposed to leverage shared eQTL across tissues to enhance the power of TWAS [43]. Alternatively, employing tissue-specific references tailored to each disease may also yield more effective results.

## Materials and methods

### TWAS fine-mapping of binary traits

We consider a genome-wide association study (GWAS) performed on $n$ individuals for a binary trait of interest. We denote **y** as an $n$-vector of binary phenotypes measured on the $n$ individuals. Our goal is to identify genes whose genetically regulated expression (GReX) is associated with the outcome trait through TWAS fine-mapping. Different from the existing TWAS fine-mapping methods [4,6], we aim to jointly model all genes on the same chromosome while explicitly accounting for the binary nature of the outcome trait. To do so, we examine one chromosome at a time. For the chromosome of focus, we denote **G** as an $n \times p$ GReX matrix constructed for $p$ genes on the chromosome for the same set of individuals. These GReX are assumed to have already been constructed using standard software such as PrediXcan [2]. With the constructed GReX, we consider a probit model linking the GReX to the binary outcome

$$P(y_i = 1|\hat{\boldsymbol{g}}_i, \boldsymbol{\alpha}) = 1 - P(y_i = 0|\hat{\boldsymbol{g}}_i, \boldsymbol{\alpha}) = \Phi(\mu + \hat{\boldsymbol{g}}_i^T \boldsymbol{\alpha})(i = 1, \cdots, n), \tag{1}$$

where $y_i$ is the binary trait for the $i'$th individual; $\hat{g}_i$ is a $p$-vector of GReX for $p$ genes in the $i'$th individual; $\alpha$ is a $p$-vector of corresponding gene effect sizes; $\mu$ is a scalar representing the intercept; and $\Phi$ is the cumulative distribution function (CDF) of the standard normal distribution. Following [44], we introduce a $p$-vector of auxiliary variables $z$ and obtain the equivalent latent variable representation of Eq [1] as:

$$y_i = \begin{cases} 1 \ if \ z_i > 0 \\ 0 \ if \ z_i \leq 0 \end{cases}, \tag{2}$$

$$z_i = \mu + \hat{g}_i^T \alpha + \epsilon_i, \epsilon_i \sim N(0, 1), \tag{3}$$

where the residual error $\epsilon_i$ follows the standard normal distribution; and $z_i$ is the $i'$th element of $z$.

Note that the GReX are included as covariates, with the weights in the linear combination of GreX ($\alpha$) automatically inferred from the model to account for their correlations. As a result, the pair-wise GReX correlation matrix is effectively accounted for in the model. To enable TWAS fine-mapping, we specify a sparsity inducing prior on each element of the gene effect sizes $\alpha$ using a point-normal distribution

$$\alpha_j \sim \pi N(0, \sigma^2) + (1 - \pi)\delta_0, \tag{4}$$

where $\alpha_j$ is the $j'$th element of $\alpha$; $\pi$ is the proportion of non-zero $\alpha$; and $\delta_0$ denotes a point mass at zero. The above equation effectively assumes that, with probability $\pi$, the $j'$th gene has a non-zero effect and its effect size follows a normal distribution with mean zero and variance $\sigma^2$. With probability $1-\pi$, the $j'$th gene has zero effect. Among the two hyper-parameters in the sparsity inducing prior, $\pi$ controls the proportion of non-zero $\alpha$ values in Eq [3] while $\sigma^2$ controls the expected magnitude of the non-zero $\alpha$ values. Specifically, for $\pi$, we assume a uniform prior distribution on $log \, \pi$ by default, with an alternative option of a beta prior on $\pi$. The prior specifications for $\pi$ and $\sigma^2$ follow that of [45] and are described in detail in the S1 Text.

## Posterior inclusion probability of genes

Based on the modeling specification described above, we developed a Markov chain Monte Carlo (MCMC) algorithm to estimate the model parameters and obtain test statistic for each gene effect size $\alpha_j$. To do so, we first introduce a set of binary indicator variables ($\gamma_j$) to represent whether the $j'$th gene has non-zero effects or not. We then apply a Metropolis-Hastings algorithm within the MCMC framework to obtain posterior samples of the hyperparameters including $\gamma_j$. To improve the convergence of the MCMC algorithm, we implement a technique known as "small world proposal" [46], where we introduce multiple local moves instead of a single move to induce longer-range proposals on the proposal distributions. The number of compounded local moves is uniformly drawn from a uniform distribution ranging from 1 to 20. Full details of the sampling algorithm are described in the S1 Text. In the present study, we used a total of 26,000 iterations, with the first 6,000 as the burn-in and the rest as the sampling iterations. With the MCMC algorithm, we obtain for each gene its posterior probability of having a non-zero effect, or $\gamma_j = 1$, a quantity commonly referred to as the posterior inclusion probability (PIP). This PIP serves as an important measure of evidence for the gene's association with the outcome trait. We refer to our method as FABIO (Fine-mApping of causal genes for BInary Outcomes).

## Simulations

We conducted simulations to assess the performance of FABIO and compare it with existing fine-mapping methods FOCUS and FOGS. To do so, we used real genotype data and simulated both gene expression and the binary outcome trait. Specifically, we randomly sampled 50,000 individuals of European ancestry in UK Biobank to serve as the GWAS data. We used all 373 individuals of European ancestry in GEUVADIS to serve as the eQTL mapping data (more details on UK Biobank and GEUVADIS are provided in the next section). We focused on chr 1 for our simulations, obtained 405,875 SNPs on chr 1 that are shared between the GWAS and eQTL data, and partitioned these SNPs into 133 independent LD blocks defined by LDetect [47], following the same procedure of [4,6]. These LD blocks contain 1–60 genes per block (mean = 12, median = 8). We retained 88 LD blocks with at least 5 genes and randomly selected 50 LD blocks among them to carry out the simulations. For the selected LD blocks, we obtained the genes inside them (5–45 genes per block, mean = 16, median = 13) and extracted a common set of cis-SNPs that reside within 100 kb upstream of the transcription start site (TSS) and 100 kb downstream of the transcription end site (TES) of each gene following [41,42]. We kept 770 genes that had at least 10 cis-SNPs and used these SNPs and genes for simulations. These genes contain a medium of 364 SNPs (average = 370) per gene, with over 95% of them having more than 100 SNPs.

We simulated the gene expression in both the GWAS and eQTL datasets following the strategy of [42]. Specifically, for each gene in turn, we randomly selected $k$ SNPs to have non-zero effects on gene expression, and then simulated its gene expression level as:

$$g_i = \boldsymbol{m}_i^T \boldsymbol{\beta} + \boldsymbol{\varepsilon}_i, \boldsymbol{\varepsilon}_i \sim N(0, 1 - PVE_1), \tag{5}$$

$$\beta_j \sim N(0, PVE_1/k). \tag{6}$$

where $g_i$ is the simulated gene expression level for the $i'$th individual (in either the GWAS data or eQTL data), $\boldsymbol{m}_i$ is a $k$-vector of standardized genotypes for the non-zero effect SNPs for the $i'$th individual, $\boldsymbol{\beta}$ is a $k$-vector of simulated SNP effect sizes with $\beta_j$ being the $j'$th element, and $PVE_1$ is a scalar that represents the proportion of gene expression variance explained by genetic effects. We will describe the choice of $k$ in a following paragraph.

We then randomly selected $n_g$ genes to have non-causal effects on the trait and simulated a binary trait based on the simulated gene expression in the GWAS panel using a logistic regression model:

$$P\{y_i = 1|z_i\} = \frac{1}{1 + exp(-z_i)}, \tag{7}$$

$$z_i = u + \boldsymbol{g}_i^T \boldsymbol{\alpha} + \epsilon_i, \epsilon_i \sim N(0, 1 - (n_g \cdot PVE_2)). \tag{8}$$

where $y_i$ is the simulated binary trait for the $i'$th individual; $z_i$ is the latent variable for the corresponding individual; $u$ is an intercept that controls the case/control ratio as it equals to the expectation of $log\ (case\#/control\#)$; $\boldsymbol{g}_i$ is a $n_g$-vector of simulated gene expression of the selected causal genes; $n_g$ is the number of selected causal genes; $\boldsymbol{\alpha}$ is a $n_g$-vector of genes' causal effects. Following [42], we set the non-zero effect $\alpha_j = \sqrt{PVE_2/PVE_1}$ for every element of $\boldsymbol{\alpha}$, where $PVE_2$ is a scalar that denotes the proportion of variance in $z_i$ that is explained by the causal genes. Note that we used a logistic regression instead of a probit model to evaluate the performance of our method under model misspecifications. We will describe the choice of $n_g$ in the following paragraph.

In the simulations, we first examined the calibration of test statistics from different methods under two null simulation settings: one is the complete null setting, where we set $PVE_2 = 0\%$ so that none of the genes were associated with the trait; the other is a masked null setting, where we randomly selected 10 LD blocks, randomly assigned one gene in each block to be causal, and masked the causal genes by excluding them in the fine-mapping analysis. Besides null simulations, we also examined the power of different methods on detecting the causal genes under the alternative simulation settings with non-zero $PVE_2$. For both complete null and alternative simulations, we started from a baseline simulation setting and then varied one parameter at a time on top of the baseline setting to examine the influence of different parameters. Specifically, for $PVE_1$, we set it to be 1%, 5%, 10% (baseline setting), or 20% following the same settings of [4]. For the number of SNPs that display non-zero effects on gene expression ($k$ in Eq [6]), we set it to be either 1, 2, 1% (baseline setting), or 10% of all cis-SNPs following [4]. For $PVE_2$, we varied its value to be either 0% (complete null setting), 0.2%, 0.4% (baseline setting), or 0.6% following [42]. For $n_g$, we followed FOCUS and assumed one causal gene in 10 of the 50 LD blocks, with no causal genes in the remaining 40 blocks. In addition, we noticed that the main simulations in FOGS paper assumed one causal gene per LD block, including a non-sparse case of all the cis-SNPs to be causal for each causal gene. Therefore, we considered four distinct settings covering both non-sparse and sparse conditions: one causal gene in 10 of the 50 LD blocks with either 1% or 100% cis-SNPs to be causal; or one causal gene in each of the 50 LD blocks with either 1% or 100% cis-SNPs to be causal, and all other parameters followed the baseline settings. For the binary trait, we examined different case:control ratios by varying the intercept $u$, using 1:1 as the baseline setting ($u = 0$) while exploring other ratios of 1:10 ($u = log (0.1)$), 1:20 ($u = log (0.05)$), and 1:50 ($u = log (0.02)$).

FABIO places a uniform prior on log $\pi$ by default. Additionally, it allows the users to assume $\pi$ following a beta prior distribution: $\pi \sim Beta(a,b)$, and input $a$ and $b$ based on their prior knowledge to control the expectation and variance of $\pi$. Here, we compared the performance of FABIO using default prior against the beta prior on $\pi$ with different combinations of $a$ and $b$ in the simulations under the baseline setting. When applying FABIO with the beta prior on $\pi$, we first tried three combinations with fixed expectation and different variances: (1) $a = 1.1$ and $b = 99$; (2) $a = 0.11$ and $b = 9.9$; (3) $a = 11$ and $b = 990$, all of which leads to the same prior expectation $E[\pi] = 1.1/100$, which is close to the true $\pi$ in our baseline setting: 10/770. Furthermore, we included two additional parameter combinations to test the influence of prior expectation on method performance: (4) $a = 1.1$ and $b = 9.9$; (5) $a = 1.1$ and $b = 999$, which either over-estimated or under-estimated proportion of causal genes as compared to the truth, respectively.

We performed 100 simulation replicates for each simulation setting. In each replicate, we applied FOCUS and FOGS to analyze one LD block at a time and applied FABIO to either analyze one LD block at a time or analyze all 50 LD blocks jointly. We then summarized the results across simulation replicates. In the analysis, we first examined the calibration of test statistics from different methods under the null simulation settings. For the frequentist method FOGS, we examined the number of false discoveries at the Bonferroni adjusted p-value threshold. For the other two Bayesian methods FABIO and FOCUS, we examined the number of false discoveries using the 95% credible set based on PIPs. Given those methods produce different test statistics, we evaluated the power of different methods using two rank-based criteria: (a) the power under a false discovery rate (FDR) of 0.05; (b) the area under the curve (AUC) based on the receiver operating characteristics (ROC) curves. Since we knew the ground truth in the simulations, we first compared the power of different methods under a true FDR threshold of 0.05. Specifically, to compute the true FDR in the simulations, we first divided 100 simulation replicates into five groups, with 20 replicates per group. In each group, we obtained the test

statistics from each method for all genes across the 20 replicates, either in the form of p-values or PIPs. For each method in turn, we sorted the test statistics from the most significant (i.e. the smallest p-value or the largest PIP) to the least significant (i.e. the largest p-value or the smallest PIP). For each ordered test statistic in the sorted list, we counted the number of false positives detected above the threshold defined by this specific test statistic. The number of false positives, divided by the total number of genes called significant at that threshold, is FDR. We then determined threshold that corresponds to an FDR of 0.05, based on which we further calculated power. Power is calculated as the number of true positives above the threshold in the sorted list divided by the total number of causal genes. Afterwards, we obtained the average power across five groups to minimize stochasticity [9,42,48–49].

Because we can only compute the true FDR in the simulations but not in the real data, we also compared the power of different methods under an estimated FDR threshold of 0.05. We estimated FDR using different approaches: the Benjamini-Hochberg approach [50] to estimate FDR using p-values from the frequentist methods, and the local FDR approach [51] to estimate FDR using PIPs from the Bayesian methods. Specifically for the local FDR approach, we first calculated the local FDR for all the genes, which is effectively 1-PIP [9,52]. We then sorted genes based on these local FDRs from small to large. For each gene in turn, we estimated FDR for that gene by aggregating the local FDRs from the first gene up to that gene in the sorted list. Afterwards, we identified the cutoff value of PIP/local FDR that corresponds to a targeted FDR threshold of 0.05. This is the significant threshold we use that corresponds to an estimated FDR of 0.05.

## Compared methods

We compared FABIO with two existing TWAS fine-mapping methods FOCUS (Version 0.6.10) and FOGS (Version 2.0), and two recently published methods GIFT (Version 1.0.0) and cTWAS (Version 0.1.37). FOCUS and cTWAS are Bayesian methods that use a two-stage regression framework for fine-mapping. Both of them first obtain the SNP prediction weights on gene expression from the eQTL mapping study, and then plugs in the estimated SNP weights on expression in GWAS summary statistics for TWAS fine-mapping. FOGS and GIFT are frequentist fine-mapping method that use a two-stage regression framework. FOGS relies on a SNP set test to examine the joint effects of all SNPs within a gene while controlling for the SNP effects of the other genes in the same region; while GIFT examines one genomic region at a time, jointly models the GReX of all genes residing in the focal region, and carries out TWAS conditional analysis in a maximum likelihood framework. In practice, we noticed that both GIFT and cTWAS are computationally demanding: it takes about three days for either method to analyze on a relatively small LD block with <10 genes and over a week on a moderate-sized LD block with >20 genes in the simulations. This is in direct contrast to the other methods: for example, it took FABIO, FOCUS and FOGS an average of 3.1, 1.5 and 9.2 hours to analyze all 50 LD blocks in the simulations, respectively. Due to the heavy computational demand of these two methods, we limited comparisons to the baseline simulation setting, and further down-sampled to 5,000 GWAS individuals to speed up GIFT, as we used individual-level GWAS data for GIFT following the developers' advice to achieve the best performance.

We used the Bayesian sparse linear mixed models (BSLMM) [13] to construct gene expression prediction models for FABIO, FOCUS and GIFT, while we used the elastic net regression [10] for FOGS following the recommendation from its developer to improve its performance, also for cTWAS following the developers' suggestions that cTWAS performs best when prediction models are sparse, and the computing time will increase dramatically for cTWAS with increasing density of variants. With the SNP weights, we directly constructed GReX in the

GWAS data for FABIO and GIFT, then carried out TWAS fine-mapping analysis. For the other three methods, we performed marginal association analysis in the GWAS data with logistic regression using PLINK 2.0 [53] to first obtain GWAS summary statistics. We then paired the GWAS summary statistics with the SNP weights as inputs for them. For those methods, we calculated the LD matrix using the same set of individuals in UKBB.

### Real datasets

**GEUVADIS data.** The GEUVADIS data [54] contain gene expression measurements for 462 individuals from five different populations that include CEPH (CEU), Finns (FIN), British (GBR), Toscani (TSI), and Yoruba. We combined samples from the four European ancestries (CEU, FIN, GBR, and TSI) and obtained a final sample size of 373. In the expression data, we only focused on protein-coding genes annotated from GENCODE [55] (release 12), and removed lowly expressed genes that had zero counts in at least half of the individuals to obtain a total of 14,698 protein-coding genes. We performed PEER [56] normalization to remove confounding effects. Following [57] to remove potential population stratification, we first quantile-normalized the gene expression measurements across individuals in each of the four European populations to a standard normal distribution, and then further quantile-normalized the expression measurements to a standard normal distribution across all 373 individuals from all four European populations. In addition to expression data, we also obtained genotype data of the 373 individuals from the 1000 Genomes Project phase 3. We filtered out SNPs that have a Hardy-Weinberg equilibrium (HWE) p-value $< 10^{-4}$, a genotype call rate $< 95\%$, or a minor allele frequency (MAF) $< 0.05$. We retained a final set of 5,395,690 SNPs for analysis.

**UK Biobank (UKBB) data.** The UK Biobank data consists of 487,409 individuals and 92,693,895 imputed autosome SNPs [58]. We followed the same sample QC procedure as Neale Lab (https://github.com/Nealelab/UK_Biobank_GWAS/tree/master/imputed-v2-gwas) to retain a total of 337,198 individuals of European ancestry. We performed SNP QC by filtering out SNPs with a minor allele frequency (MAF) $< 0.01$, with an imputation information score $< 0.8$, and with a Hardy-Weinberg equilibrium (HWE) test p-value $< 10^{-10}$. After these QC steps, we retained 6,725,312 SNPs for our analysis.

### Real data application

We performed fine-mapping on six binary disease traits through integrating the GEUVADIS data with the UKBB data. The examined disease traits include asthma, breast cancer, gout, hypertension, prostate cancer, and rheumatoid arthritis. These disease traits were selected following [59] and have a wide range of prevalence (0.02 to 0.38). For the two sex-related traits (breast cancer and prostate cancer), we limited our analysis to female individuals (for breast cancer) or male individuals (for prostate cancer). In the analysis, we extracted cis-SNPs that resided within 100 kb upstream of TSS and 100 kb downstream of TES of the gene and focused on a common set of cis-SNPs among them that were shared between UKBB and GEUVADIS datasets. We retained a total of 14,388 genes that contained at least 10 cis-SNPs for TWAS fine-mapping analysis. For a given trait in UKBB, we kept all individuals of European ancestry with self-reported case status of that trait in the GWAS data. Same as the simulations, we used BSLMM to construct GReX prediction models for FABIO, FOCUS, and GIFT, and used the elastic net regression for FOGS and cTWAS following the recommendation from its developer. For FABIO, we used individual-level genotypes of the GWAS data and analyzed genes on each chromosome jointly. To correct for population structure and relatedness, we first identified the top 10 principal components (PCs) of genotypes using principal component analysis (PCA). We then regressed the predicted expression level of each gene on the top 10

genotype PCs and sex, and used the linear regression residual as the corrected gene expression level as input for FABIO. We applied FABIO using both the default uniform prior on log π, and the alternative beta prior on π. For the alternative beta prior, we fixed the parameter *a* of the beta prior to be 1.1, and estimated the parameter *b* for each disease trait by assuming one causal gene per risk region that contains GWAS or TWAS signals (defined in Table 1). For the other four methods, we first performed marginal association analysis in the GWAS data with logistic regression model using PLINK 2.0 [53] including the top 10 genotype PCs and sex as covariates. We then paired the GWAS summary statistics with the SNP weights as input, and analyzed 1,433 independent LD blocks [47] across all 22 autosomal chromosomes one at a time. For GIFT and cTWAS, due to the heavy computational demand mentioned earlier in the simulations, we again restricted the comparison to a subset of LD blocks that contain no more than 20 genes, covering a total of 8,411 genes from 1,259 unique LD blocks. We compared the results of these two methods with genes identified by FABIO, FOCUS, or FOGS from the same subset of the LD blocks. We also performed marginal TWAS analysis using FUSION [1] with the GWAS summary statistics and the SNP weights calculated by BSLMM. For a fair comparison across methods, we used an estimated FDR threshold of 0.05 for all methods to declare significance. We estimated FDR based on p-values using the Benjamini-Hochberg approach for FOGS and GIFT, and estimated FDR based on PIPs using the local FDR approach for FABIO, FOCUS, and cTWAS. For the significant genes identified by FABIO, FOCUS, and FOGS, we further performed enrichment analysis with the Gene Ontology and Kyoto Encyclopedia of Genes and Genomes (KEGG) pathways using the clusterprofiler R package [60]. We declared gene set significance based on a *q*-value threshold of 0.05.

## Software overview

FABIO is implemented in an R package and is freely available at https://github.com/superggbond/FABIO/. FABIO requires an individual-level predicted GReX file of the TWAS cohort of study (generated using standard methods like PrediXcan and BSLMM) in either plain text format or gzipped text format, which has *m* (number of genes) rows and *n* (sample size) + 1 columns. The gene names serve as the first column, following *n* columns where each column represents GReX of genes for an individual. If the users only have genotypes from an individual-level GWAS in PLINK format, FABIO also provides a function to create the predicted GReX file using our prebuilt BSLMM gene expression prediction models derived from the GEUVADIS eQTL panel, the same eQTL panel used in our real data applications. FABIO also requires the binary phenotypes for the same cohort of *n* individuals in a plain text format, using 1 to indicate the case and 0 to indicate the control. The order of the binary phenotypes should be consistent with the column order of the predicted GReX file. A key parameter in the model is π, which represents the proportion of genes with non-zero effects. We present two prior options for π: a uniform prior on log π, which requires no input of π from the users, and the initial π will be the proportion of genes passing the single-gene tests based on the input predicted GReX file; or a beta prior $\pi \sim Beta(a,b)$, where users can choose the values of a and b if they have prior knowledge to determine the prior expectation and the variance of π. At the end, FABIO outputs the TWAS fine-mapping results as a table with three columns: the names of the genes, the PIPs of the genes, and the FDR estimate for each gene corresponding to the PIP value.

## Supporting information

**S1 Table. Summary results of TWAS fine-mapping in UK Biobank (FABIO input options for π).** The table summarizes the number of discoveries for each of the six disease traits (rows) in the TWAS fine-mapping analysis of UK Biobank. A risk region with GWAS or TWAS

signals (1st column) is defined as an LD block that harbors at least one genome-wide significant SNP or significant TWAS gene same as the definition we applied in Table 1 of the main text. The following two columns list the number of genes discovered by applying FABIO with default prior and beta prior on $\pi$, respectively. The number in the bracket is the number of identified genes that are located in a risk region with GWAS or TWAS signals. We used an estimated FDR threshold of 0.05 to declare significance in the fine-mapping analysis. (DOCX)

**S2 Table. Summary results of TWAS fine-mapping in UK Biobank (five methods).** The table summarizes the number of discoveries for each of the six disease traits (rows) in the subset TWAS fine-mapping analysis of UK Biobank. We covered 8,411 genes from 1,259 unique LD blocks that contain no more than 20 genes in each LD block. A risk region with GWAS or TWAS signals (1st column) is defined as an LD block that harbors at least one genome-wide significant SNP or significant TWAS gene same as the definition we applied in Table 1 of the main text. The following five columns list the number of genes discovered by each of the five methods. The number in the bracket is the number of identified genes that are located in a risk region with GWAS or TWAS signals. We used an estimated FDR threshold of 0.05 to declare significance for all methods in the fine-mapping analysis. (DOCX)

**S3 Table. Summary results of TWAS fine-mapping in UK Biobank (down-sampled).** The table summarizes the number of discoveries for each of the six disease traits (rows) in the reduced sample TWAS fine-mapping analysis of UK Biobank. A GWAS risk region (1st column) is defined as an LD block that harbors at least one genome-wide significant SNP (p-value $< 5 \times 10^{-8}$). A significant TWAS gene (2nd column) is defined as a gene with a marginal TWAS p-value $< 0.05/14,388$. A TWAS risk region (3rd column) is defined as an LD block that harbors at least one marginal TWAS significant gene. A risk region with GWAS or TWAS signals (4th column) is defined as an LD block that harbors at least one genome-wide significant SNP or significant TWAS gene. The last three columns list the number of genes discovered by each of the three methods. The number in the bracket is the number of identified genes that are located in a risk region with GWAS or TWAS signals. We used an estimated FDR threshold of 0.05 to declare significance for all methods in the fine-mapping analysis. (DOCX)

**S1 Fig. Performance of different methods in controlling false signals. (a)** Number of false signal genes per simulation replicate under different complete null settings (the first 10 settings) and the masked gene null setting (the last setting). **(b)** Quantile-quantile plot of -log10 p-values for testing the non-causal genes using FOGS under the complete null simulation setting. **(c)** Quantile-quantile plot of -log10 p-values for testing the non-causal genes using FOGS under the masked null simulation setting. **(d)** Comparisons among different methods for the estimate of false discovery rate (FDR) under the alternative baseline simulation setting. Compared methods include FABIO (salmon pink), FOCUS (green) and FOGS (blue). The estimated FDR (y-axis) is plotted against the true FDR (x-axis). (TIF)

**S2 Fig. True FDR at an estimated FDR threshold of 0.05 under different conditions.** The red dashed line indicates a true FDR of 0.05. (a) Under different numbers of causal SNP(s). (b) Under different percentages of PVE2. (c) Under different sparsity of causal genes and SNPs. (TIF)

**S3 Fig. Performance of different methods under the baseline simulation setting. (a)** Power comparison for different methods based on a true false discovery rate (FDR) of 0.05. FABIO was applied to either analyze all LD blocks jointly (FABIO-chr) or one LD block at a time (FABIO-LD). FOCUS and FOGS were only applied to analyze one LD block at a time. **(b)** ROC curves of different methods with AUCs recorded in the plot.
(TIF)

**S4 Fig. Performance of different methods in simulations with the change of the number of causal SNP(s).** (a) Average number of genes in 95% credible set (CS) defined by FABIO or FOCUS and the number of true signal genes in 95% CS. (b) Power comparison for different methods based on a true false discovery rate (FDR) of 0.05. (c) ROC curves of different methods with AUCs recorded.
(TIF)

**S5 Fig. Performance of different methods in simulations with the change of PVE2.** (a) Average number of genes in 95% credible set (CS) defined by FABIO or FOCUS and the number of true signal genes in 95% CS. (b) Power comparison for different methods based on a true false discovery rate (FDR) of 0.05. (c) ROC curves of different methods with AUCs recorded.
(TIF)

**S6 Fig. Performance of different methods in simulations with the change of sparsity of causal genes and SNPs.** (a) Average number of genes in 95% credible set (CS) defined by FABIO or FOCUS and the number of true signal genes in 95% CS. (b) Power comparison for different methods based on a true false discovery rate (FDR) of 0.05. (c) ROC curves of different methods with AUCs recorded.
(TIF)

**S7 Fig. Power comparison for different methods based on an estimated false discovery rate (FDR) threshold of 0.05.** (a) under different case/control ratios. (b) under different PVE1. (c) under different numbers of causal SNP(s). (d) under different PVE2. (e) under different sparsity of causal genes and SNPs.
(TIF)

**S8 Fig. Performance of FOCUS and FOGS with GWAS summary statistics generated by either logistic regression or linear regression.** (a) Power comparison based on a true false discovery rate (FDR) threshold of 0.05. (b) ROC curves with AUCs recorded.
(TIF)

**S9 Fig. Performance of FABIO with default or beta prior on π.** (a) Power comparison based on a true false discovery rate (FDR) threshold of 0.05. (b) ROC curves with AUCs recorded.
(TIF)

**S10 Fig. Performance of all five methods. (a)** Power comparison based on a true false discovery rate (FDR) threshold of 0.05. **(b)** ROC curves with AUCs recorded.
(TIF)

**S11 Fig. TWAS Manhattan plot of asthma.** Genes identified in TWAS fine-mapping analysis are labeled in black boxes. Each gene is represented as a dot with x-axis indicating its genomic location and y-axis indicating its -log10 of p-value in the marginal TWAS association test. The dot of each gene is then colored based on different categories: (1) only identified by FABIO; (2) only identified by FOCUS; (3) only identified by FOGS; (4) identified by both FABIO and FOCUS; (5) identified by both FABIO and FOGS; (6) identified by both FOCUS and FOGS;

(7) identified by all three methods; (8) located in a known risk region but missed by all three methods (uncaptured signal); (9) a gene shown no significance in both TWAS and GWAS analyses (non-signal).
(TIF)

**S12 Fig. Results of the Gene Ontology and KEGG pathway enrichment analysis.** Dot plots show the significant terms of identified genes identified by different fine-mapping methods, with color gradients representing statistical significance based on q-value. (**a**) Gene Ontology enriched pathways for asthma using significant genes identified by FABIO. (**b**) KEGG enriched pathways for asthma using significant genes identified by FABIO. (**c**) KEGG enriched pathways for asthma using significant genes identified by FOCUS. (**d**) Gene Ontology enriched pathways for hypertension using significant genes identified by FABIO. (**e**) KEGG enriched pathways for hypertension using significant genes identified by FABIO. (**f**) Gene Ontology enriched pathways for hypertension using significant genes identified by FOCUS. (**g**) KEGG enriched pathways for hypertension using significant genes identified by FOCUS. (**h**) Gene Ontology enriched pathways for hypertension using significant genes identified by FOGS.
(TIF)

**S13 Fig. Box plot of pairwise correlation of GReX for four distinct groups of gene pairs.** (1) Within LD: within each LD block, (2) Nearby LD: between two adjacent LD blocks, (3) Other LD: between two distant LD blocks, and (4) Btw chr LD: between one LD block on chromosome 1 and one on chromosome 2.
(TIF)

**S1 Text. Supplementary text for the methods.**
(DOCX)

## Author Contributions

**Conceptualization:** Kevin He, Lam C. Tsoi, Xiang Zhou.

**Data curation:** Haihan Zhang, Zheng Li.

**Formal analysis:** Haihan Zhang, Zheng Li.

**Funding acquisition:** Lam C. Tsoi, Xiang Zhou.

**Methodology:** Haihan Zhang, Xiang Zhou.

**Software:** Haihan Zhang.

**Supervision:** Kevin He, Lam C. Tsoi, Xiang Zhou.

**Validation:** Haihan Zhang.

**Visualization:** Haihan Zhang.

**Writing – original draft:** Haihan Zhang.

**Writing – review & editing:** Haihan Zhang, Kevin He, Lam C. Tsoi, Xiang Zhou.

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
