## [Decision Letter · Decision Letter 0]

9 Apr 2024

Dear Dr Zhang,

Thank you very much for submitting your Research Article entitled 'FABIO: TWAS Fine-mapping to Prioritize Causal Genes for Binary Traits' to PLOS Genetics.

The manuscript was fully evaluated at the editorial level and by independent peer reviewers. The reviewers appreciated the attention to an important problem, but raised some substantial concerns about the current manuscript. Based on the reviews, we will not be able to accept this version of the manuscript, but we would be willing to review a much-revised version. We cannot, of course, promise publication at that time.

If you decide to revise the manuscript for further consideration at PLOS Genetics, please aim to resubmit within the next 60 days, unless it will take extra time to address the concerns of the reviewers, in which case we would appreciate an expected resubmission date by email to plosgenetics@plos.org.

We are sorry that we cannot be more positive about your manuscript at this stage. Please do not hesitate to contact us if you have any concerns or questions.

Yours sincerely,

Xiaofeng Zhu

Section Editor

PLOS Genetics

Xiaofeng Zhu

Section Editor

PLOS Genetics

Reviewer's Responses to Questions

**Comments to the Authors:**

Reviewer #1: In this manuscript, you present FABIO, a novel multivariate TWAS method aimed at identifying disease-associated genes. FABIO offers two key advantages over existing methods. First, it is specifically designed for binary traits using individual-level data. Second, it can simultaneously analyze all genes on a chromosome, rather than focusing on a single gene or multiple genes within a single cis-region, as is the case with current methods. I highly commend the methodological innovation of this work and anticipate that it will play a significant role in future TWAS analyses. However, there are some issues with the presentation of the manuscript, and I have a few questions that I hope the authors can address or clarify.

Treating binary traits as continuous in existing methods?

To my knowledge, existing methods such as FOCUS and FOGS, which are based on GWAS summary data, do not approximate binary variables as continuous ones. In fact, for binary traits, GWAS tests the marginal correlations between a variant and the liability score of this binary trait (i.e., the latent variable z_i in FABIO, equations 2 and 3). While there may be bias due to model misspecification (i.e., the generating model is a liability model (probit model) while the estimation model is a logistic model), this is not the same issue as the one claimed by the authors in the second paragraph of the introduction. The authors also state in their simulations that these existing methods are based on GWAS summary data, rather than performing a linear model on binary outcomes. As mentioned in the General Comment, I have no doubt about the methodological innovation of FABIO, but it does not address these specific issues with existing methods.

Fairness of simulation for FOGS

I observed unusual results in Figures 2 and 3: FOGS has a very high type-I error (based on FDR) and extremely low power. After reading the authors' processing methods, I found that they used simulations to discover the potential type-I error of each method (2.6% for FABIO, 3.3% for FOCUS, 46.5% for FOGS) and then used these thresholds to calculate power. This is unfair to FOGS. After reading about FOGS, I discovered that it is not based on a sparsity assumption, and it is normal for its type-I error to be higher than that of FABIO and FOCUS, which utilize a sparsity assumption. The authors then use this very high type-I error as the threshold for determining power, which exaggerates the shortcomings of FOGS. In addition, the terminology "true FDR of 0.05" is somewhat misleading. I believe that the analysis of type-I error and power should be independent, and the threshold should be chosen using the same method (e.g., q-value < 0.05). The phrase "simulated threshold of 0.05 FDR" may be more appropriate and fair.

Sufficiency of simulation replications

As other reviewers may also point out, the number of simulation replications is only 20, which may not be sufficient for evaluating these three methods. I believe 100 replications should be performed for the main cases (e.g., those shown in Figures 2 and 3).

Addressing GReX correlation across the chromosome?

FABIO utilizes traditional GReX modeling, and the process of predicting SNP weights only uses LD information from the cis-region. In other words, although gene expression may have many variants with strong trans-regulatory effects (trans-eQTL), this trans-eQTL information is not utilized in predicting gene expression, resulting in LD-wise uncorrelated predicted gene expression. Therefore, my initial view is that although FABIO processes all genes on a chromosome simultaneously, it does not leverage GReX correlations among genes between LD blocks. Although these predicted genes are independent, analyzing them in the same model can utilize prior distribution information to control FDR, which I believe is an advantage of FABIO.

Optimal tuning parameter selection

FABIO is a variable selection method, which belongs to the same category as lasso and SuSiE. For lasso and SuSiE, the penalty parameter lambda and the number of single effects L are the main tuning parameters that control their performance, respectively. In FABIO, the proportion of non-zero effects pi is a crucial tuning parameter. In the supplementary materials, the authors claim that they assume log pi follows a uniform distribution on U(log(1/p),0), while traditional methods tend to assume pi follows a beta distribution, adjusting a and b (the two parameters of the beta distribution) to control the mean and variance of pi. Can the performance of FABIO be improved by modifying the prior distribution of pi if the users have prior knowledge about the approximate proportion of important genes on a chromosome?

Software availability and documentation

I noticed that the software for FABIO has been made publicly available with a relatively detailed tutorial. I suggest that the authors prominently highlight this point in the manuscript (https://superggbond.github.io/FABIO/). I also suggest adding a short paragraph somewhere to briefly introduce how to use it, what inputs are, and which (important) tuning parameters are recommended to specify manually.

Reviewer #2: In the manuscript under consideration, the authors presented a novel method, FABIO, using a Bayesian additive model to fine map genes in a TWAS framework. The mathematical formulation of the method is clearly presented. Simulations and real data analyses are professionally done. Although I think it is a great piece of work, there are some aspects may need to be clarified and explained more intuitively. Here are my comments for the authors’ considerations.

Major:

First, I feel that the idea of using a linear combination of GReX to predict phenotype for fine mapping a little bit counter intuitive. The default reason why fine mapping is needed is that the genes are correlated to each other, therefore better modeling of such correlations are required. However, the model just adds all genes (GReX) up, despite the potential problem of collinearity. Intuitively, the models that explicitly taking correlation structure (e.g., the pair-wise correlation matrix, analogue to the LD matrix of all variants in GWAS fine-mapping) into account may be the right solution. I didn’t mean the presenting method is total wrong. At least some intuitive interpretation is needed here.

Second, there is a lack of details on how the cutoff of the methods are set to ensure a fair comparison between the three methods. It has been repetitively stated that FDR = 0.05 is used without specifics on how this FDR is calculated. However, I can’t figure out a default way to calculate FDR in the present framework with PIP. This is an important procedure that has not been presented.

Minor:

The explanation of Figure 1 and the first few paragraphs in Results are redundant to existing texts in Methods. It appears to me that the paper was originally prepared in a format that Results show up earlier leaving Methods to the end. Better flow aligning to PLoS Genetics may be required.

The comparison to other methods could be more thorough. For instance, there are two newer papers published in Nat Genet on this topic (including the one published from the same group of this paper). It may be nice to carry out some comparisons.

Reviewer #3: This study presents a novel method to prioritize causal genes by integrating GWAS and eQTL studies. It is novel in two aspects. First, it is designed for binary traits with a probit model. Second, it accounts for all genes on one chromosome. The authors evaluated the performance of the new method (FABIO) with simulated date and compared its performance to two existing methods (FUSION and FOGS). The performances of the three methods were also compared in the applications to six binary traits in the UK Biobank dataset. This manuscript is very well-written. It is very easy to follow, including the methodological details. With various simulation scenarios (e.g., case/control ratio, phenotypic variance explained, number of causal SNPs, and number of causal genes), the authors presented convincing evidence that FABIO has well-controlled false positive rates and much better power than the other two methods. In the application to real data, FABIO narrowed down to the smallest number of genes while identifying genes in the likely risk regions, defined based on GWAS or TWAS significant signals. The findings from FABIO are more robust to down-sampling. Gene set and pathway enrichment analysis further supported the biological relevance of the candidate genes fine-mapped by FABIO. Lastly, FABIO has acceptable computational efficiency for biobank-scale dataset. Overall, this study developed an outstanding method and package for the field of GWAS. It is likely to be widely used in GWAS of binary traits. It is quite a pleasure to read this manuscript, which is informative and easy to follow. I support the publication of the manuscript in its current form. I do have one suggestion for the author’s consideration.

• The very poor performance of one of the comparison method, FOGS, in both simulation and application is kind of surprising. It will be useful to readers if the authors could discuss or propose some explanations. Does it have to do with the specific simulation approaches? Or with the binary traits, instead of quantitative traits?

**Have all data underlying the figures and results presented in the manuscript been provided?**

Reviewer #1: Yes

Reviewer #2: Yes

Reviewer #3: Yes

PLOS authors have the option to publish the peer review history of their article (what does this mean?). If published, this will include your full peer review and any attached files.

Reviewer #1: No

Reviewer #2: No

Reviewer #3: No

---

## [Decision Letter · Decision Letter 1]

9 Nov 2024

PGENETICS-D-24-00208R1FABIO: TWAS Fine-mapping to Prioritize Causal Genes for Binary TraitsPLOS Genetics Dear Dr. Zhang, Thank you for submitting your manuscript to PLOS Genetics. As you can see, both reviewer 2 and 3  have no further questions and agree the manuscript is acceptable.  However, reviewer 1 still has some minor questions. Therefore, we invite you to submit a revised version of the manuscript that addresses reviewer 1's questions accordingly.  Please submit your revised manuscript within 30 days Dec 09 2024 11:59PM. If you will need more time than this to complete your revisions, please reply to this message or contact the journal office at plosgenetics@plos.org. Please include the following items when submitting your revised manuscript:*
A rebuttal letter that responds to each point raised by the editor and reviewer(s). You should upload this letter as a separate file labeled 'Response to Reviewers'. This file does not need to include responses to formatting updates and technical items listed in the 'Journal Requirements' section below.*
A marked-up copy of your manuscript that highlights changes made to the original version. You should upload this as a separate file labeled 'Revised Manuscript with Track Changes'.*
An unmarked version of your revised paper without tracked changes. You should upload this as a separate file labeled 'Manuscript'. If you would like to make changes to your financial disclosure, competing interests statement, or data availability statement, please make these updates within the submission form at the time of resubmission. Guidelines for resubmitting your figure files are available below the reviewer comments at the end of this letter. We look forward to receiving your revised manuscript. Kind regards, Xiaofeng ZhuSection EditorPLOS Genetics Xiaofeng ZhuSection EditorPLOS Genetics Aimée DudleyEditor-in-ChiefPLOS Genetics Anne GorielyEditor-in-ChiefPLOS Genetics **Journal Requirements:** **Additional Editor Comments (if provided):****Reviewers' comments:** Reviewer's Responses to Questions

**Comments to the Authors:**

Reviewer #1: I am pleased and grateful that you have provided a comprehensive response, and have modified your method using one of my suggestions. I believe you have appropriately incorporated the changes.

However, I still have a minor question regarding the calculation of the true FDR and local FDR. I am grateful the you can address, as they may be due to my limited knowledge in this area.

1. When calculating the true FDR in the simulations, why did the you choose to divide the simulations into 5 groups?

2. I am very interested in your approach to adjusting the PIP based on Efron's method. This could potentially benefit all current work based on SuSiE. After reading Efron's paper, I understand that this method requires estimating the density function of a mixture model and pre-specifying some tuning parameters like p0 (equation 2.13). While Efron provided some guidance on how to estimate these, I was hoping you could share a more detailed pipeline on how you applied this method in your work (e.g., whether it is able to leverage any existing R packages to facilitate the estimation process).

3. I would like you to help me understand that the term "local" in the local FDR context refers to a statistical concept, and not a genetic locus or a block in the block-wise GReX estimation.

Reviewer #2: The authors have addressed my previous comments satisfactorily.

Reviewer #3: The authors did an impressive amount of work to revise the manuscript and address my and other reviewers' concerns. The current manuscript is of publication quality. This research will have significant impact in the field of GWAS.

**Have all data underlying the figures and results presented in the manuscript been provided?**

Reviewer #1: Yes

Reviewer #2: Yes

Reviewer #3: Yes

PLOS authors have the option to publish the peer review history of their article (what does this mean?). If published, this will include your full peer review and any attached files.

Reviewer #1: No

Reviewer #2: No

Reviewer #3: **Yes: **Kaixiong Ye

 **Figure resubmission:** While revising your submission, please upload your figure files to the Preflight Analysis and Conversion Engine (PACE) digital diagnostic tool, https://pacev2.apexcovantage.com/. PACE helps ensure that figures meet PLOS requirements. To use PACE, you must first register as a user. Registration is free. Then, login and navigate to the UPLOAD tab, where you will find detailed instructions on how to use the tool. If you encounter any issues or have any questions when using PACE, please email PLOS at figures@plos.org. Please note that Supporting Information files do not need this step. If there are other versions of figure files still present in your submission file inventory at resubmission, please replace them with the PACE-processed versions. **Reproducibility:** To enhance the reproducibility of your results, we recommend that authors deposit laboratory protocols in protocols.io, where a protocol can be assigned its own identifier (DOI) such that it can be cited independently in the future. Additionally, PLOS ONE offers an option to publish peer-reviewed clinical study protocols. Read more information on sharing protocols at https://plos.org/protocols?utm_medium=editorial-email&utm_source=authorletters&utm_campaign=protocols

---

## [Editor Report · Decision Letter 2]

14 Nov 2024

Dear Dr Zhang,

We are pleased to inform you that your manuscript entitled "FABIO: TWAS Fine-mapping to Prioritize Causal Genes for Binary Traits" has been editorially accepted for publication in PLOS Genetics. Congratulations!

Yours sincerely,

Xiaofeng Zhu

Section Editor

PLOS Genetics

Xiaofeng Zhu

Section Editor

PLOS Genetics

Aimée Dudley

Editor-in-Chief

PLOS Genetics

Anne Goriely

Editor-in-Chief

PLOS Genetics

Comments from the reviewers (if applicable):

**Data Deposition**

http://datadryad.org/submit?journalID=pgenetics&manu=PGENETICS-D-24-00208R2

**Press Queries**

---

## [Editor Report · Acceptance letter]

25 Nov 2024

PGENETICS-D-24-00208R2 

FABIO: TWAS Fine-mapping to Prioritize Causal Genes for Binary Traits 

Dear Dr Zhang, 

We are pleased to inform you that your manuscript entitled "FABIO: TWAS Fine-mapping to Prioritize Causal Genes for Binary Traits" has been formally accepted for publication in PLOS Genetics! Your manuscript is now with our production department and you will be notified of the publication date in due course.

With kind regards,

Zsofia Freund

PLOS Genetics

On behalf of:
